# THE PRINCIPLE OF LOGIT SEPARATION

## ABSTRACT

We consider neural network training, in applications in which there are many possible classes, but at test-time, the task is to identify only whether the given example belongs to a specific class, which can be different in different applications of the classifier. For instance, this is the case in an image search engine. We consider the *Single Logit Classification* (SLC) task: training the network so that at test-time, it would be possible to accurately identify if the example belongs to a given class, based only on the output logit for this class. We propose a natural principle, the *Principle of Logit Separation*, as a guideline for choosing and designing losses suitable for the SLC. We show that the cross-entropy loss function is not aligned with the Principle of Logit Separation. In contrast, there are known loss functions, as well as novel batch loss functions that we propose, which are aligned with this principle. In total, we study seven loss functions. Our experiments show that indeed in almost all cases, losses that are aligned with Principle of Logit Separation obtain a 20%-35% relative performance improvement in the SLC task, compared to losses that are not aligned with it. We therefore conclude that the Principle of Logit Separation sheds light on an important property of the most common loss functions used by neural network classifiers.

`Tensorflow` code for optimizing the new batch losses will be made publicly available upon publication; A URL will be provided in the publication version of this manuscript.

## 1 INTRODUCTION

With the advent of Big Data, classifiers can learn fine-grained distinctions, and are used for classification in settings with very large numbers of classes. Datasets with up to hundreds of thousands of classes are already in use in the industry (Deng et al., 2009; Partalas et al., 2015), and such classification tasks have been studied in several works (e.g., Weston et al. 2013; Gupta et al. 2014). Classification with a large number of classes appears naturally in vision, in language modeling and in machine translation (Bahdanau et al., 2015; Józefowicz et al., 2016; Dean et al., 2013).

When using neural network classifiers, one implication of a large number of classes is a high computational burden at test-time. Indeed, in standard neural networks using a softmax layer and the cross-entropy loss, the computation needed for finding the logits of the classes (the pre-normalized outputs of the top network layer) is linear in the number of classes (Grave et al., 2017), and can be prohibitively slow for high-load systems, such as search engines and real-time machine translation systems.

In many applications, the task at test-time is not full classification of each example into one of the many possible classes. Instead, the task, at each application of the classifier, is to identify whether the example should be classified into one of a small subset of the possible classes, or even a single class. This class can be different in different applications of the classifier. For instance, for face recognition, we might train a classifier to classify a large number of faces, but at test-time, in each execution we need to search for photos of one specific person. Therefore we do not need to identify the person in each photo — we only need to identify whether this is a photo of the person of interest. Another example is neural machine translation systems, where vocabulary sizes can reach hundreds of thousands of words. In this case, by allowing fast evaluation of the posterior probabilities of a subset of the classes, one could implement a fast decoder for translations. This decoder would first try a small subset that includes the most frequent words, and only if a suitable translation was not found, it would try the entire vocabulary.

In this type of applications, one would ideally like to have a test-time computation that does not depend on the total number of possible classes. A natural approach is to calculate only the logit of the class of interest, and use this value alone to infer whether this is the true class of the example. However, the logit of a single class might be meaningful only in comparison to logits of other classes, in which case unless the other logits are also calculated, it cannot be used for successfully determining whether the example belongs to the class of interest. We name the goal of inferring class correctness from the logit of this class alone *Single Logit Classification* (SLC).

In this work, we show that when using the standard cross-entropy loss for training, the value of a single logit is not informative enough for determining whether this is indeed the true class for the example. In other words, the cross-entropy loss yields poor performance in the SLC task. Further, we identify a simple principle that we name the *Principle of Logit Separation*. This principle captures an essential property that a loss function must have in order to yield good performance in the SLC task. The principle states that to succeed in the SLC task, the training objective should optimize for the following property:

> *The value of any logit that belongs to the correct class of any training example should be larger than the value of any logit that belongs to a wrong class of any (same or other) training example.*

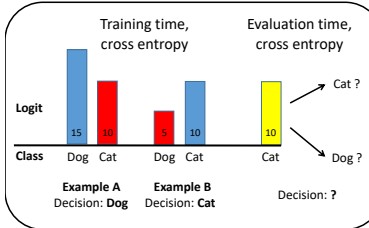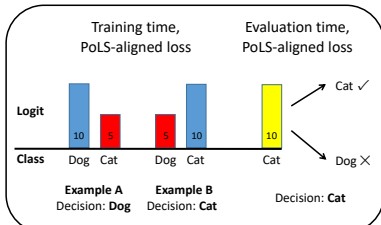

Figure 1: The Principle of Logit Separation. Left: when training with the cross-entropy loss, the logit values for the class 'Cat' can be the same for two examples, one where it is the true class (blue) and one where it is not (red). Therefore, at test-time, a logit with the same value for the class 'Cat' does not indicate whether the example belongs to this class. Right: With a loss function that is aligned with the Principle of Logit Separation, all true logits are greater than all false logits at training time. Hence, at test time, a single logit can indicate the correctness of its respective class.

We give a formal definition of the Principle of Logit Separation in Section 2. See Figure 1 for an illustration. We study previously suggested loss functions and their alignment with the Principle of Logit Separation. We show that the Principle of Logit Separation is satisfied by the self-normalization (Devlin et al., 2014) and Noise-Contrastive Estimation (Mnih & Teh, 2012) training objectives, proposed for calculating posterior distributions in the context of natural language processing, as well as by the binary cross-entropy loss used in multi-label settings (Wang et al., 2016; Huang et al., 2013). In contrast, the principle is not satisfied by the standard cross-entropy loss and by the max-margin loss. We derive new training objectives for the SLC task based on the Principle of Logit Separation. These objectives are novel batch versions of the cross-entropy loss and the max-margin loss, and we show that they are aligned with the Principle of Logit Separation. In total, we study seven different training objectives.

We corroborate in experiments that the Principle of Logit Separation indeed explains the difference in performance of the different loss functions in the SLC task, concluding that training with a loss function that is aligned with the Principle of Logit Separation results in logits that are significantly more informative as a standalone value. Specifically, in almost all cases that we tested, training with a loss function that is aligned with the Principle of Logit Separation achieved a 20%-35% relative performance improvement in the SLC task, compared to the loss functions that are not aligned with this principle, such as the cross-entropy loss. Moreover, the performance in the SLC task of losses that are aligned with the Principle of Logit Separation was usually better even than the performance of the normalized cross-entropy logits, despite the fact that the normalized logits are calculated using the values of the logits of all available classes, a computationally demanding task.

Our main contributions are the following:

- Introducing the Principle of Logit Separation, a simple principle that sheds light on a property of the most common loss functions used by neural network classifiers.

- Analyzing the alignment of seven losses with the Principle of Logit Separation. These losses include two novel losses that we propose. `Tensorflow` code for optimizing the novel loss functions will be made publicly available upon publication.

- Showing that the Principle of Logit Separation indeed explains the success of different loss functions in the SLC task. Specifically, objectives that satisfy the Principle of Logit Separation outperform standard objectives on the SLC task with a 20%-35% relative performance improvement in almost all cases, while keeping multiclass classification accuracy the same or higher. Furthermore, these objectives outperform even the normalized logits derived from the cross-entropy loss, in which case the logits of all classes are computed.

We conclude that the Principle of Logit Separation is an important and valuable property for neural network training objectives, when using single logit values for test-time classification as in the SLC task.

**Related Work**    We review existing methods that are relevant for faster test-time classification. The hierarchical softmax layer (Morin & Bengio, 2005; Mnih & Hinton, 2008) replaces the flat softmax layer with a binary tree with classes as leaves, making the computational complexity of calculating the posterior probability of each class logarithmic in the number of classes. A drawback of this method is the additional construction of the binary tree of classes, which requires expert knowledge or data-driven methods. Inspired by the hierarchical softmax approach, Grave et al. (2017) exploit unbalanced word distributions to form clusters that explicitly minimize the average time for computing the posterior probabilities over the classes. The authors report an impressive speed-up factor of between 2 and 10 for posterior probability computation, but their computation time still depends on the total number of classes. Differentiated softmax was introduced in Chen et al. (2016) as a less computationally expensive alternative to the standard softmax mechanism, in the context of neural language models. With differentiated softmax, each class (word) is represented in the last hidden layer using a different dimensionality, with higher dimensions for more frequent classes. This allows a faster computation for less frequent classes. However, this method is applicable only for highly unbalanced class distributions. Several sampling-based approaches were developed in the context of language modeling, with the goal of approximating the softmax function at training-time. Notable examples are importance sampling (Bengio & Senecal, 2003; 2008), negative sampling (Mikolov et al., 2013), and Noise Contrastive Estimation (NCE) (Gutmann & Hyvärinen, 2010; Mnih & Teh, 2012). These methods do not necessarily improve the test-time computational burden, however we show below that the NCE loss can be used for the SLC task.

## 2    THE PRINCIPLE OF LOGIT SEPARATION

In the SLC task, the only information about an example is the output logit of the model for the single class of interest. Therefore, a natural approach to classifying whether the class matches the example is to set a threshold: if the logit is above the threshold, classify the example as belonging to this class, otherwise, classify it as not belonging to the class. We refer to logits that belong to the true classes of their respective training examples as *true logits* and to other logits as *false logits*. For the threshold approach to work well, the values of all true logits should be larger than the value of all false logits across the training sample (in fact, it is enough to separate true and false logits on a class level, but we stick to the stronger assumption in this work). This is illustrated in Figure 1. The Principle of Logit Separation (PoLS), which was stated in words in Section 1, captures this requirement. We formalize this principle below.

Let $[k] := \{1, \ldots, k\}$ be the possible class labels. Assume that the training sample is $S = ((x_1, y_1), \ldots, (x_n, y_n))$, where $x_i \in \mathbb{R}^d$ are the training examples, and $y_i \in [k]$ are the labels of these examples. For a neural network model parametrized by $\theta$, we denote by $z_y^\theta(x)$ the value of the logit assigned by the model to example $x$ for class $y$. The Principle of Logit Separation (PoLS) can be formally stated as follows:

**Definition 2.1** (The Principle of Logit Separation). *The* Principle of Logit Separation *holds for a labeled set $S$ and a model $\theta$, if for any $(x, y), (x', y') \in S$ (including the case $x = x', y = y'$) and any $y'' \neq y'$, we have $z_y^\theta(x) > z_{y''}^\theta(x')$.*

The definition assures that every true logit $z_y^\theta(x)$ is larger than every false logit $z_{y''}^\theta(x')$. If this simple principle holds for all train and test examples, it guarantees perfect accuracy in the SLC task, since all true logits are larger than all false logits. Thus, a good approach for a training objective for SLC is to attempt to optimize for this principle on the training set. For a loss $\ell$, $\ell(S, \theta)$ is the value of the loss on the training sample using model $\theta$. A loss $\ell$ is aligned with the Principle of Logit Separation if for any training sample $S$, a small enough value of $\ell(S, \theta)$ ensures that the requirement in Definition 2.1 is satisfied for the model $\theta$. In the following sections we study the alignment with the PoLS of known losses and new losses.

## 3 STANDARD OBJECTIVES IN VIEW OF THE PoLS

In this section we show that the cross-entropy loss (Hinton, 1989), which is the standard loss function for neural network classifiers (e.g., Krizhevsky et al. 2012) and the multiclass max-margin loss (Crammer & Singer, 2001), do not satisfy the PoLS.

**The cross-entropy loss**    The cross-entropy loss on a single example is defined as

$$\ell(z, y) = -\log(p_y), \quad \text{where} \quad p_y := e^{z_y} / \sum_{j=1}^{k} e^{z_j} = \Big( \sum_{j=1}^{k} e^{z_j - z_y} \Big)^{-1}. \tag{1}$$

Note that $p_y$ is the probability assigned by the softmax layer. It is easy to see that the cross-entropy loss does not satisfy the PoLS. Indeed, as the loss depends only on the difference between logits for every example separately, minimizing it guarantees a certain difference between the true and false logits for every example separately, but does not guarantee that all true logits are larger than all false logits in the training set. Formally, the following counter-example shows that this loss is not aligned with the PoLS. Let $S = ((x_1, 1), (x_2, 2))$ be the training sample, and let $\theta_\alpha$, for $\alpha > 0$, be a model such that $z^{\theta_\alpha}(x_1) = (2\alpha, \alpha)$, and $z^{\theta_\alpha}(x_2) = (-2\alpha, -\alpha)$. Then $\ell(S_{\theta_\alpha}) = 2\log(1 + e^{-\alpha})$. Therefore for any $\epsilon > 0$, there is some $\alpha > 0$ such that $\ell(S_{\theta_\alpha}) \leq \epsilon$, but $z_2^{\theta_\alpha}(x_1) > z_2^{\theta_\alpha}(x_2)$, contradicting an alignment with PoLS.

**The max-margin loss**    Max-margin training objectives, most widely known for their role in training Support Vector Machines, are used in some cases for training neural networks (Tang, 2013; Socher et al., 2011; Janocha & Czarnecki, 2017). Here we consider the multiclass max-margin loss suggested by Crammer & Singer (2001), defined as

$$\ell(z, y) = \max(0, \gamma - z_y + \max_{j \neq y} z_j), \tag{2}$$

where $\gamma > 0$ is a hyperparameter that controls the separation margin between the true logit and the false logits of the example. It is easy to see that this loss too does not satisfy the PoLS, since minimizing it again guarantees only a certain difference between the true and false logits for every example separately, and not across the entire training sample. Indeed, consider the same training sample $S$ as defined in the counter-example for the cross-entropy loss above, and the model $\theta_\alpha$ defined there. Setting $\alpha = \gamma$, we have $\ell(S_{\theta_\gamma}) = 0$. Thus for any $\epsilon > 0$, $\ell(S_{\theta_\gamma}) < \epsilon$, but $z_2^{\theta_\gamma}(x_1) > z_2^{\theta_\gamma}(x_2)$, contradicting an alignment with PoLS.

## 4 OBJECTIVES THAT SATISFY THE PoLS

In this section we consider objectives that have been previously suggested for addressing problems that are somewhat related to the SLC task. We show that these objectives indeed satisfy the PoLS.

### 4.1 SELF-NORMALIZATION

Self-normalization (Devlin et al., 2014) was introduced in the context of neural language models, to avoid the costly step of computing the posterior probability distribution over the entire vocabulary when evaluating the trained models. The self-normalization loss is a sum of the cross-entropy loss

with an additional term. Let $\alpha > 0$ be a hyperparameter, and $p_y$ as defined in Eq. (1). The self-normalization loss is defined by

$$\ell(z,y) = -\log(p_y) + \alpha \cdot \log^2(\sum_{j=1}^{k} e^{z_j}).$$

The motivation for this loss is self-normalization: The second term is minimal when the softmax normalization term $\sum_{j=1}^{k} e^{z_j}$ is equal to 1. When it is equal to 1, the exponentiated logit $e^{z_j}$ can be interpreted as the probability that the true class for the example is $j$. Devlin et al. (2014) report a speed-up by a factor of 15 in evaluating models trained when using this loss, since the self-normalization enables computing the posterior probabilities for only a subset of the vocabulary.

Intuitively, this loss should also be useful for the SLC task: If the softmax normalization term is always close to 1, there should be no need to compute it, thus only the logit of the class in question should be required to infer whether this class in the correct one for the example. Indeed, we show that the self-normalization loss is aligned with the PoLS. When the first term in the loss is minimized for an example, correct and wrong logits are as different as possible from one another. When the second term is minimized for an example, the sum of exponent logits is equal to one. Therefore, when both terms are minimized for an example, the correct logit converges to zero while wrong logits converge to negative infinity. When this is done for the whole training sample, all correct logits are larger than all wrong logits in the training sample. A formal proof is provided in Appendix A.1.

## 4.2 NOISE CONTRASTIVE ESTIMATION

Noise Contrastive Estimation (NCE) (Gutmann & Hyvärinen, 2010; Mnih & Teh, 2012) was considered, like self-normalization, in the context of natural language learning. This approach was developed to speed up neural-language model training over large vocabularies. In NCE, the multi-class classification problem is treated as a set of binary classification problems, one for each class. Each binary problem classifies, given a context and a word, whether this word is from the data distribution or from a noise distribution. Using only $t$ words from the noise distribution (where $t$ is an integer hyperparameter) instead of the entire vocabulary leads to a significant speedup at training-time. Similarly to the self-normalization objective, NCE, in the version appearing in Mnih & Teh (2012), is known to produce a self-normalized logit vector (Andreas & Klein, 2015). This property makes NCE a good candidate for the SLC task, as single logit values are informative for the class correctness, and not only when compared other logits in the same example.

The loss function used in NCE for a single training example, as given by Mnih & Teh (2012), is defined based on a distribution over the possible classes, denoted by $q = (q(1), \ldots, q(k))$, where $\sum_{i=1}^{k} q(i) = 1$. The NCE loss, in our notation, is

$$\ell(z,y) = -\log g_y - t \cdot \mathbb{E}_{j\sim q}\left[\log(1 - g_j)\right], \text{ where } g_j := (1 + t \cdot q(j) \cdot e^{-z_j})^{-1}, \qquad (3)$$

During training, the second term in the loss is usually approximated by Monte-Carlo approximation, using $t$ random samples of $j \sim q$, to speed up training time (Mnih & Teh, 2012).

We observe that NCE loss is aligned with the PoLS. First, observe that $g_j$ is of a similar form to $\sigma(z_j)$ where $\sigma(z) = (1 + e^{-z})^{-1}$ is the sigmoid function. Therefore, it is easy to see that when the term above is minimized for one example, the value of true logit $z_y$ converges to infinity, and the values of all false logits converge to negative infinity. When the above term is minimized for the entire training set, all true logits are larger than all false logits across the training set. A formal proof is provided in Appendix A.2.

## 4.3 BINARY CROSS-ENTROPY

The last known loss that we consider is often used in multilabel classification settings. In multilabel settings, each example can belong to several classes, and the goal is to identify the set of classes an example belongs to. A common approach (Wang et al., 2016; Huang et al., 2013) is to try to solve $k$ binary classification problems of the form "Does $x$ belong to class $j$?" using a single neural network model, by minimizing the sum of the cross-entropy losses that correspond to these binary problems. In this setting, the label of each example is a binary vector $(r_1, \ldots, r_k)$, where $r_j = 1$ if $x$ belongs

to class $j$ and 0 otherwise. The loss for a single training example with logits $z$ and label-vector $r$ is

$$\ell(z, (r_1, \ldots, r_k)) = -\sum_{j=1}^{n} r_j \log(\sigma(z_j)) + (1 - r_j)\log(1 - \sigma(z_j)),$$

Where $\sigma(z) = (1 + e^{-z})^{-1}$ is the sigmoid function. This loss can also be used for our setting of multiclass problems, by defining $r_j := \mathbf{1}_{j=y}$ for an example $(x, y)$. This gives the multiclass loss

$$\ell(z, y) = -\log(\sigma(z_y)) + \sum_{j \neq y} \log(1 - \sigma(z_j)).$$

The binary cross-entropy is also aligned with the PoLS. Indeed, similarly to case of the NCE loss, it is easy to see that when the term above is minimized for one example, the value of true logit $z_y$ converges to infinity, and the values of all false logits converge to negative infinity. When the above term is minimized for the entire training set, all true logits are larger than all false logits across the training set. A formal proof is provided in Appendix A.3.

## 5 NEW TRAINING OBJECTIVES FOR THE SLC TASK

In this section we propose new training objectives for the SLC task, designed to satisfy the PoLS. These objectives adapt the training objectives of cross-entropy and max-margin, studied in Section 3, that do not satisfy the PoLS, by generalizing them to optimize over *batches* of training samples. We show that the revised losses satisfy the PoLS. This approach does not require any new hyper-parameters, since the batch size is already a hyperparameter in standard Stochastic Gradient Descent. Further, this allows an easy adaptation of available neural network implementations to the SLC task. When the cross-entropy loss or the max-maring loss are minimized, they guarantee a certain difference between the true and the false logits of each example separately. Our generalization of these losses to batches of examples enforces an ordering also between true and false logits of different examples.

### 5.1 BATCH CROSS-ENTROPY

Our first batch loss generalizes the cross-entropy loss, which was defined in Eq. (1). The cross-entropy loss can be given as the Kullback-Leibler (KL) divergence between two distributions, as follows. The KL divergence between two discrete probability distributions $P$ and $Q$ over $[k]$ is defined as $\mathrm{KL}(P||Q) := \sum_{i=j}^{k} P(j)\log(P(j)/Q(j))$. For an example $(x, y)$, let $P_{(x,y)}$ be the distribution over $[k]$ which deterministically outputs $y$, and let $Q_x$ be the distribution defined by the softmax normalized logits, $Q_x(j) = e^{z_j} / \sum_{i=1}^{k} e^{z_i}$. Then it is easy to see that for $p_y$ as defined in Eq. (1), $\mathrm{KL}(P_{(x,y)}||Q_x) = -\log p_y$, exactly the cross-entropy loss in Eq. (1).

We define a batch version of this loss, using the KL-divergence between distributions over batches. Recall that the $i$'th example in a batch $B$ is denoted $(x_i, y_i)$. Let $P_B$ be the distribution over $[m] \times [k]$ defined by

$$P_B(i, j) := \begin{cases} \frac{1}{m} & j = y_i, \\ 0 & \text{otherwise.} \end{cases}$$

Let $Q_B$ be the distribution defined by the softmax normalized logits over the entire batch $B$. Formally, denote $Z(B) := \sum_{i=1}^{m} \sum_{j=1}^{k} e^{z_j(x_i)}$. Then $Q_B(i, j) := e^{z_j(x_i)}/Z(B)$. We then define the batch cross-entropy loss as follows.

**Definition 5.1** (The batch cross-entropy loss). *Let $m > 1$ be an integer, and let $B$ be a uniformly random batch of size $m$ from $S$. The* batch cross-entropy loss *of a training sample $S$ is*

$$\ell(S) := \mathbb{E}_B[L_c(B)], \quad where \quad L_c(B) := \mathrm{KL}(P_B||Q_B).$$

This batch version of the cross-entropy loss is aligned with the PoLS. Indeed, when this loss is minimized for one training batch, all true logits converge to some positive value (as a normalized exponentiated true logit converges to $1/m$), while all false logits converge to negative infinity (as

a normalized exponentiated false logit converges to zero). Therefore, when minimizing this loss across the whole training set, all true logits are larger than all false logits in the training set. A formal proof is provided in Appendix A.4.

## 5.2 BATCH MAX-MARGIN.

Our second objective is a batch version of the max-margin loss, which was defined in Eq. (2). For a batch $B$, denote the minimal true logit in $B$, and the maximal false logit in $B$, as follows:

$$z_+^B := \min_{(x,y) \in B} z_y(x), \quad \text{and} \quad z_-^B := \max_{(x,y) \in B, j \neq y} z_j(x).$$

**Definition 5.2** (The batch max-margin loss). *Let $m > 1$ be an integer, and let $B$ be a uniformly random batch of size $m$ from $S$. Let $\ell$ be the single-example max-margin loss defined in Eq. (2), let $\gamma > 0$ be the max-margin hyper-parameter. The* batch max-margin *is defined by*

$$\ell(S) := \mathbb{E}_B[L_m(B)], \quad \text{where} \quad L_m(B) := \frac{1}{m} \max(0, \gamma - z_+^B + z_-^B) + \frac{1}{m} \sum_{(x,y) \in B} \ell(z(x), y).$$

The batch version of the max-margin loss is aligned with the PoLS. Minimizing the first term in the loss makes sure that all true logits in the batch are larger than all false logits in the batch. Therefore, minimizing the loss over the entire training set makes sure that the PoLS holds. A formal proof is provided in Appendix A.4. Note that while the seconds term in the loss is not necessary for ensuring alignment with the PoLS, it is necessary for practical reasons, as without it the gradient is propagated through only two logits from the entire minibatch, which leads to harder optimization and poorer generalization.

## 6 EXPERIMENTS

We compared the performance of neural networks trained with each of the objectives studied above, on the SLC task and on multiclass classification. To evaluate a learned model on the SLC task, for each class $j$ and a threshold $T$, we measured the precision and recall in identifying examples from class $j$ using the test $z_j > T$, and calculated the Area Under the Precision-Recall curve (AUPRC) defined by the entire range of possible thresholds. We also measured the precision at fixed recall values 0.9 (Precision@0.9) and 0.99 (Precision@0.99). We report the averages of these values over all the classes in the dataset. We further report the multiclass accuracy (Acc.) of each model.

We evaluated the methods on five computer-vision classification benchmark datasets: MNIST (Le-Cun et al., 1998), SVHN (Netzer et al., 2011) CIFAR-10 and CIFAR-100 (Krizhevsky & Hinton, 2009). The last dataset is Imagenet (Russakovsky et al., 2015), which has 1000 classes, demonstrating the scalability of the PoLS approach to many classes. Due to its size, training on Imagenet is highly computationally intensive, therefore we evaluated its performance using two representative methods, which do not require tuning additional hyperparameters. For every dataset, a single standard network architecture was used for all training objectives.

The network architectures we used are standard, and were fixed before running the experiments. For the MNIST dataset, we used an MLP comprised of two fully-connected layers with 500 units each, and an output layer, whose values are the logits, with 10 units. For the SVHN, CIFAR-10 and CIFAR-100 datasets, we used a convolutional neural network (LeCun et al., 1989) with six convolutional layers and one dense layer with 1024 units. The first, third and fifth convolutional layers used a $5 \times 5$ kernel, where other convolutional layers used a $1 \times 1$ kernel. The first two convolutional layers were comprised of 128 feature maps, where convolutional layers three and four had 256 feature maps, and convolutional layers five and six had 512 feature maps. Max-pooling layers with $3 \times 3$ kernel size and a $2 \times 2$ stride were applied after the second, fourth and sixth convolutional layers. In all networks, batch normalization (Ioffe & Szegedy, 2015) was applied to the output of every fully-connected or convolutional layer, followed by a rectified-linear non-linearity. For every combination of a training objective and a dataset (with its fixed network architecture), we optimized for the best learning rate among $1, 0.1, 0.01, 0.001$ using the classification accuracy on a validation set. Except for Imagenet, each model was trained for $10^5$ steps, which always sufficed for convergence. For the Imagenet experiments, we used an inception-v3 architecture (Szegedy et al., 2016) as appears in the

| Dataset | Method | 1-AUPRC | 1-Precision@0.9 | 1-Precision@0.99 | 1-Acc. |
|---|---|---|---|---|---|
| MNIST | CE | 0.008 | 0.005 | 0.203 | **0.012** |
| | max-margin | 0.012 | 0.018 | 0.262 | 0.014 |
| | self-norm | 0.002 | 0.001 | **0.021** | 0.014 |
| | NCE | 0.002 | 0.002 | **0.021** | 0.013 |
| | binary CE | 0.002 | **0.000** | 0.037 | 0.014 |
| | batch CE | **0.001** | 0.001 | 0.022 | 0.013 |
| | batch max-margin | 0.002 | 0.001 | 0.034 | 0.013 |
| | CE with all logits | 0.001 | 0.000 | 0.020 | 0.012 |
| SVHN | CE | 0.023 | 0.028 | 0.545 | 0.044 |
| | max-margin | 0.021 | 0.025 | 0.532 | 0.043 |
| | self-norm | **0.015** | 0.014 | 0.298 | **0.039** |
| | NCE | 0.021 | 0.017 | 0.320 | 0.042 |
| | binary CE | **0.015** | 0.016 | 0.312 | 0.041 |
| | batch CE | **0.015** | **0.013** | **0.280** | **0.039** |
| | batch max-margin | 0.018 | 0.020 | 0.384 | 0.047 |
| | CE with all logits | 0.015 | 0.016 | 0.313 | 0.044 |
| CIFAR-10 | CE | 0.109 | 0.326 | 0.703 | 0.146 |
| | max-margin | 0.094 | 0.285 | 0.705 | 0.145 |
| | self-norm | 0.073 | 0.204 | 0.599 | 0.139 |
| | NCE | 0.081 | 0.214 | **0.594** | 0.143 |
| | binary CE | **0.070** | 0.210 | 0.607 | **0.137** |
| | batch CE | 0.072 | **0.202** | 0.602 | 0.140 |
| | batch max-margin | 0.075 | 0.226 | 0.636 | 0.147 |
| | CE with all logits | 0.074 | 0.214 | 0.648 | 0.146 |
| CIFAR-100 | CE | 0.484 | 0.866 | 0.974 | 0.416 |
| | max-margin | 0.490 | 0.893 | 0.977 | 0.466 |
| | self-norm | 0.378 | 0.807 | 0.970 | 0.401 |
| | NCE | 0.383 | **0.795** | 0.964 | 0.415 |
| | binary CE | 0.426 | 0.870 | 0.978 | 0.445 |
| | batch CE | **0.371** | **0.795** | **0.961** | **0.400** |
| | batch max-margin | 0.468 | 0.903 | 0.983 | 0.473 |
| | CE with all logits | 0.380 | 0.801 | 0.973 | 0.416 |
| Imagenet (1000 classes) ($6 \cdot 10^6$ iterations) | CE | 0.366 | 0.739 | 0.932 | 0.286 |
| | batch CE | **0.245** | **0.563** | **0.865** | **0.278** |
| | CE with all logits | 0.223 | 0.566 | 0.872 | 0.286 |

Table 1: Results on Single Logit classification, using the different loss functions. In almost all cases, loss functions that are aligned with the Principle of Logit Separation (under the dashed line) yield a relative improvement of 20%-35% in the different performance measures, while also yielding a small improvement in classification performance.

`tensorflow` (Abadi et al., 2015) repository. We used all the default hyperparameters from this implementation, changing only the loss function used. For every tested loss function, we trained the inception-v3 model for $6 \cdot 10^6$ iterations.

Experiment results are reported in Table 1. Since many of the measures in our experiments are close to the maximal value of 1, we report the value of *one minus* each measure, so that a smaller number indicates a better accuracy. For each dataset, the losses above the dashed line do not satisfy the PoLS (Section 3), while the losses below the line do (Sections 4 and 5). Finally, the bottom row in each dataset stands for cross-entropy with *all logits*: here we used the output logits of the cross-entropy loss *after softmax normalization*, a method which requires computing all the logits, unlike the other methods that we consider. The performance of this method on the SLC is likely close to the best possible in the SLC task, since this method uses information from all logits, unlike

the other methods that we tested. In the table, the best result for each dataset and measure, out of all the losses excluding the cross-entropy with all logits, is indicated in boldface.

Several observations can be gleaned from the results:

1. In almost all cases, all training objectives that are aligned with the PoLS yield a 20%-35% relative performance improvement in the SLC task, compared to the training objectives that are not aligned with the PoLS.

2. The objectives that are aligned with the PoLS usually also achieve a better classification accuracy in standard multiclass classification.

3. Even when all cross-entropy logits are calculated and the logits are normalized, the performance in the SLC task is comparable to the performance of the loss functions that are aligned with the PoLS, which use only a single logit.

We conclude from these experiments that indeed, alignment with the PoLS is a crucial ingredient for success in the SLC task. Further, it can be seen that the SLC task can be achieved, while keeping the accuracy on the multiclass classification task the same or higher.

## 7 CONCLUSION

In this work we considered the Single Logit Classification task, which is important in various applications. We formulated the Principle of Logit Separation, a simple principle that sheds light on an important property of the most common loss functions used by neural network classifiers. We explained, and corroborated in experiments, that a loss function that is aligned with the Principle of Logit Separation yields class logits that are significantly more informative regarding the correctness of their respective classes. In almost all cases, using these more informative logits, obtained by training a classifier with a loss function that is aligned with the Principle of Logit Separation, yielded a 20%-35% relative performance improvement in the Single Logit Classification task.

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

# A    OMMITTED PROOFS: ALIGNMENT WITH THE PRINCIPLE OF LOGIT SEPARATION

All the losses that We consider are a function of the output logits and the labels of the examples. For a neural network model $\theta$, denote the vector of logits it assigns to example $x$ by $z^\theta(x) = (z_1^\theta(x), \ldots, z_k^\theta(x))$. When $\theta$ and $x$ are clear from context, we write $z_j$ instead of $z_j^\theta(x)$. Denote the logit output of the sample by $S_\theta = ((z^\theta(x_1), y_1), \ldots, (z^\theta(x_n), y_n))$. A loss function for neural network training is a function $\ell : \cup_{n=1}^\infty (\mathbb{R}^k \times [k])^n \to \mathbb{R}_+$, which assigns a loss to a training sample based on the output logits of the model and on the labels of the training examples. The goal of training is to find a model $\theta$ which minimizes $\ell(S_\theta) \equiv \ell(S, \theta)$. In almost all the losses we study below, the loss on the training sample is simply the sum over all examples of a loss defined on a single example: $\ell(S_\theta) \equiv \sum_{i=1}^n \ell(z^\theta(x_i), y_i)$, thus it suffices to define $\ell(z, y)$. We explicitly define $\ell(S_\theta)$ below only when it deviates from this paradigm.

## A.1    SELF-NORMALIZATION

We prove that the self-normalization loss satisfies the PoLS. Assume a training sample $S$ and a neural network model $\theta$, and consider an example $(x, y) \in S$. We consider the two terms of the loss in order. First, consider $-\log(p_y)$. From the definition of $p_y$ (Eq. 1) we have that

$$-\log(p_y) = \log(\sum_{j=1}^k e^{z_j - z_y}) = \log(1 + \sum_{j \neq y} e^{z_j - z_y}).$$

Set $\epsilon_0 := \log(1 + e^{-2})$. Then, if $-\log(p_y) < \epsilon_0$, we have $\sum_{j \neq y} e^{z_j - z_y} \leq e^{-2}$, which implies that (a) $\forall j \neq y, z_j \leq z_y - 2$ and (b) $e^{z_y} \geq \sum_{j=1}^k e^{z_j}/(1 + e^{-2}) \geq \frac{1}{2} \sum_{j=1}^k e^{z_j}$. Second, consider the second term. There is an $\epsilon_1 > 0$ such that if $\log^2(\sum_{j=1}^k e^{z_j}) < \epsilon_1$ then (c) $2e^{-1} < \sum_{j=1}^k e^{z_j} < e$, which implies $e^{z_y} < e$ and hence (d) $z_y < 1$.

Now, consider $\theta$ such that $\ell(S_\theta) \leq \epsilon := \min(\epsilon_0, \epsilon_1)$. Then for every $(x, y) \in S$, $\ell(z^\theta(x), y) \leq \epsilon$. From (b) and (c), $e^{-1} < \frac{1}{2} \sum_{j=1}^k e^{z_j} < e^{z^y}$, hence $z_y > -1$. Combining with (d), it follows that $-1 < z_y < 1$. Combined with (a), it follows that for $j \neq y$, $z_j < -1$. To summarize, we have shown that for every $(x, y), (x', y') \in S$ and $y'' \neq y'$, we have that $z_y^\theta(x) > -1 > z_{y''}^\theta(x')$, implying alignment with the PoLS.

## A.2    NOISE-CONTRASTIVE ESTIMATION

Recall the definition of the NCE loss from Eq. (3):

$$\ell(z, y) = -\log g_y - t \cdot \mathbb{E}_{j \sim q} [\log(1 - g_j)], \text{ where } g_j := (1 + t \cdot q(j) \cdot e^{-z_j})^{-1}.$$

We prove that the NCE loss satisfies the PoLS. The proof relies on the observation that $g_j$ is monotonic increasing in $z_j$. Therefore, if the loss is small, $g_y$ must be large and $g_j$, for $j \neq y$, must be small. Formally, fix $t$, and assume a training sample $S$. There exists an $\epsilon_0 > 0$ such that if $-\log g_j \leq \epsilon_0$, then $z_j > 0$. In addition, there exists an $\epsilon_1 > 0$ (which depends on $q$) such that if $-\mathbb{E}_{j \sim q}[\log(1 - g_j)] \leq \epsilon_1$ then for all $j \neq y$, $\log(1 - g_j)$ must be small enough so that $z_j < 0$. Now, consider $\theta$ such that $\ell(S_\theta) \leq \epsilon := \min(\epsilon_0, \epsilon_1)$. Then for every $(x, y) \in S$, $\ell(z^\theta(x), y) \leq \epsilon$. This implies that for every $(x, y), (x', y') \in S$ and $y'' \neq y'$, we have that $z_y^\theta(x) > 0 > z_{y''}^\theta(x')$, thus this loss is aligned with the PoLS.

## A.3    BINARY CROSS-ENTROPY

It can be seen that this loss is very similar in form to the NCE loss, by noting that for $g_j$ as defined in Eq. (3), $g_j = \sigma(z_j - \ln(t \cdot q(j)))$. Since in the proof for NCE we only used the monotonicity of $g_j$ in $z_j$, which holds also for $\sigma(z_j)$, an analogous argument shows that the binary cross-entropy loss satisfies the PoLS.

## A.4   BATCH LOSSES

Recall that the batch losses are defined as $\ell(S_\theta) := \mathbb{E}_B[L(B_\theta)]$, where $B_\theta$ is a random batch out of $S_\theta$ and $L$ is $L_c$ for the cross entropy (Definition 5.1), and $L_m$ is the max-margin loss (Definition 5.2).

If true logits are greater than false logits in every batch separately when using, then the PoLS is satisfied on the whole sample, since every pair of examples appears together in some batch. The following lemma formalizes this claim:

**Lemma A.1.** *If $L$ is aligned with the PoLS, and $\ell$ is defined by $\ell(S_\theta) := \mathbb{E}_B[L(B_\theta)]$, then $\ell$ is also aligned with the PoLS.*

*Proof.* Assume a training sample $S$ and a neural network model $\theta$. Since $L$ is aligned with the PoLS, there is some $\epsilon' > 0$ such if $L(B_\theta) < \epsilon'$, then for each $(x,y), (x',y') \in B$ and $y'' \neq y'$ we have that $z_y^\theta(x) > z_{y''}^\theta(x')$. Let $\epsilon = \epsilon'/\binom{n}{m}$, and assume $\ell(S_\theta) < \epsilon$. Since there are $\binom{n}{m}$ batches of size $m$ in $S$, this implies that for every batch $B$ of size $m$, $L(B_\theta) \leq \epsilon'$. For any $(x,y), (x',y') \in S$, there is a batch $B$ which includes both examples. Therefore, for $y'' \neq y'$, $z_y^\theta(x) > z_{y''}^\theta(x')$. Since this holds for any pair of examples in $S$, $\ell$ is also aligned with the PoLS. $\square$

**Batch cross-entropy**   To show that the batch cross-entropy satisfies the PoLS, we show that $L_c$ does, which by Lemma A.1 implies this for $\ell$. By the continuity of KL, and since for discrete distributions, $\text{KL}(P||Q) = 0 \iff P \equiv Q$, there is an $\epsilon > 0$ such that if $L(B_\theta) \equiv \text{KL}(P_B||Q_B^\theta)] < \epsilon$, then for all $i, j$, $|P_B(i,j) - Q_B^\theta(i,j)| \leq \frac{1}{2m}$. Therefore, for each example $(x,y) \in B$,

$$\frac{e^{z_y^\theta(x)}}{Z(B)} > \frac{1}{2m}, \qquad \text{and} \qquad \forall j \neq y, \quad \frac{e^{z_j^\theta(x)}}{Z(B)} < \frac{1}{2m}.$$

It follows that for any two examples $(x,y), (x',y') \in B$, if $y \neq y'$, then $z_y^\theta(x) > \frac{1}{2m} > z_{y'}^\theta(x')$. Therefore $L$ satisfies the PoLS, which completes the proof.

**Batch max-margin**   To show that the batch max-margin loss satisfies the PoLS, we show this for $L_m$ and invoke Lemma A.1. Set $\epsilon = \gamma/m$. If $L(B_\theta) < \epsilon$, then $\gamma - z_+^B + z_-^B < \gamma$, implying $z_+^B > z_-^B$. Hence, any $(x,y), (x',y') \in B$ such that $y \neq y'$ satisfy $z_y^\theta(x) \geq z_+^B > z_-^B \geq z_{y'}^\theta(x')$. Thus $L$ is aligned with the PoLS, implying the same for $\ell$.

