# OpenReview forum: "The Principle of Logit Separation"
_ICLR.cc/2018/Conference — Reject_

### Official Review · AnonReviewer2 · 2017-11-26
**A good solution to the problem of speeding up test-time classification is given. More motivation for the importance of the problem is needed**

**Rating:** 6
**Confidence:** 3

**Review:**

The paper is well-written which makes it easy to understand its main
thrust - choosing loss functions so that at test time one can
accurately (and speedily) determine whether an example is in a given
class, ie loss functions which are aligned with the "Principle of Logit
Separation (PoLS)".

When the "Principle of logit separation" was first given (second page)
I found it confusing and difficult to parse (too many "any"s, I could
not work out how the quantification worked). However, the formal
definition (Definition 2.1) was fine. Why not just use this - and drop
the vague, wordy definition?

The paper is fairly 'gentle'. For example, we are taken through
examples of loss functions which satisfy "PoLS" and those which don't.
No 'deep' mathematical reasoning is required - but I don't see this as
a deficiency.

The experiments are reasonably chosen and, as expected, show the
benefits of using PoLS-aligned loss functions.

My criticism of the paper is that I don't think there is enough
motivation. We have that normal classification is linear in the number
of classes. This modest computational burden (ie just linear),
apparently, is too slow for certain applications.  I would like more
evidence for this, including some examples of this problem including
in the paper. This is lacking from the current version.


typos, etc

max-maring -> max-margin
the seconds term -> the second term

---

### Official Review · AnonReviewer3 · 2017-11-27
**Unconvincing formalization of a challenge.**

**Rating:** 3
**Confidence:** 4

**Review:**

The paper addresses the problem of a mismatch between training classification loss and a loss at test time. This is motivated by use cases in which multiclass classification problems are learned during training, but where binary or reduced multi-class classifications is performed at test time. The question for me is the following: if at test time, we have to solve "some" binary classification task, possibly drawn at random from a set of binary problems (this is not made precise in the paper), then why not optimize the same classification error or a surrogate loss at training time? Instead, the authors start with a multiclass problem, which may introduce a computational burden. when the number of classes is large as one needs to compute a properly normalized softmax. The authors now seem to ask, what if one were to use a multi-classification loss at training time, but then decides at test time that a binary classification of one-vs-all is asked for.

If one buys into the relevance of the setting, then of course, one is faced with the problem that the multiclass logits (aka raw scores) may not be calibrated to be used for binary classification by applying a fixed threshold. The authors call this sententiously "Principle of logit separation". Not too surprisingly, the standard multiclass losses do not have the desired property, however approaches that reduce multi-class to binary classification at training time do, namely unnormalized models with penalized log Z (self-normalization), the NCE approach, as well as (the natural in the proposed setting) binary classification loss. I find this almost a bit circular in the line of argumentation, but ok. It remains odd that while usually one has tried to reduce multiclass to binary, the authors go the opposite direction.

The main technical contribution of the paper is the batch-nornalization that makes sure that multiclass logits across mini-batches of data are better calibrated. One can almost think of that as an additional regularization. This seems interesting and does not create much overhead, if one applies mini-batched SGD optimization anyway. However, I feel this technique would need to be investigated with regard to general improvements in a multiclass setting and as such also benchmarked relative to other methods that could be applied.

---

> ### Author Response · Authors · 2017-12-26
> **We agree on the final conclusion, but important points are overlooked**
>
> In the setting we consider, we are presented with a multi-class classification problem, such that every example has exactly one correct class. At test time we are interested in the ability to perform fast binary (one-vs-all) classification of any given class, based on the class’s logit alone. We name this task Single Logit Classification (SLC). This setting appears naturally in the case of search engines, when the class for the binary classification is chosen according to user’s behavior. Another setting this problem applies to is the the task of face verification, where at test time we want to know if a given image is of person x of not.
>
> We agree on the final conclusion from this review, that the softmax + cross entropy training mechanism is not suited for the task we describe, and other loss functions should be used. However, we claim this conclusion and the insights we derive towards it are far from being trivial. We address specific claims below.
>
> "if at test time, we have to solve "some" binary classification task, possibly drawn at random from a set of binary problems (this is not made precise in the paper), then why not optimize the same classification error or a surrogate loss at training time?"
> In our work, we draw the same conclusion as the reviewer, that the multi-class classification should not be used in this case. However, this is far from being trivial as the reviewer suggests, for several reasons:
> 1) The problem we consider is naturally a multi-class classification problem. Among all possible classes, exactly one class is the correct one. For example, consider of classifying a face image where the classes are a large number of persons.
> 2) Several existing works do use the multiclass classification problem, instead of a more suitable loss for the problems they consider. For example, the task of face verification is often done by learning a face classifier over thousands of classes, using variants of the multi-class softmax + cross entropy mechanisms. Then, for face verification, a distance between representations of two faces in the last network layer is measured. Works that do that include:
> Parkhi, Omkar M., Andrea Vedaldi, and Andrew Zisserman. "Deep Face Recognition." BMVC. Vol. 1. No. 3. 2015.
> Liu, Weiyang, et al. "SphereFace: Deep Hypersphere Embedding for Face Recognition." arXiv preprint arXiv:1704.08063 (2017).
> Taigman, Yaniv, et al. "Deepface: Closing the gap to human-level performance in face verification." Proceedings of CVPR 2014.
> We show that using better suited loss functions may replace the need to compare face representations in the last network layer.
> 3) We outline the set of loss functions that can be used for the setting we present. Specifically, we show that suitable loss functions are ones with good logit separation characteristics.
>
> "Not too surprisingly, the standard multiclass losses do not have the desired property, however approaches that reduce multi-class to binary classification at training time do, namely unnormalized models with penalized log Z (self-normalization), the NCE approach, as well as (the natural in the proposed setting) binary classification loss"
> Again, the reviewer finds our conclusion well motivated and sound. However, we argue that this is far from being trivial, for several reasons:
> 1) Many existing works do not practice this conclusion, such as the examples listed above.
> 2) The reviewer claims that, trivially, approaches that reduce multi-class to binary classification at training time perform well in the SLC task, such as self-normalization and others. While binary cross-entropy and NCE indeed reduce multi-class to binary classification at training time, other losses we consider do not do this reduction, but still perform well on the SLC task. For example, in contrary to the reviewer's statement, self-normalization and other penalized log Z losses do not reduce multi-class to binary classification at training time. Such losses perform well on the SLC task for another reason, which is good logit separation properties, as we show in this work. In addition to self-normalization and other penalized log Z losses, we show that other losses perform well on the SLC task, such as the batch cross-entropy and batch max-margin, which also do not reduce multi-class to binary classification at training time, and again, we show that the reason for the desired behavior is the principle of logit separation.
>
> Moreover, the value of this work is by shedding light on an important property of the most common loss functions. The use of loss mechanisms such as the softmax + cross entropy, binary cross entropy and NCE is extremely abundant nowadays. Yet, basic properties about the resulting logits are still poorly understood, and this is, in our view, the most significant point of this work, which was overlooked in this review.

---

### Official Review · AnonReviewer1 · 2017-11-28
**Neat basic idea, but not enough**

**Rating:** 4
**Confidence:** 4

**Review:**

This paper explores a neat, simple idea intended to learn models suitable for fast membership queries about single classes ("is this data point a member of this class [or set of classes]?"). In the common case when the class prediction is made with a softmax function minimizing 1-of-K multiclass cross-entropy loss, this cannot in general be determined without essentially evaluating all K logits (inputs to the softmax). This paper describes how other losses (such as the natural multilabel cross-entropy) do not suffer this problem because all true labels' logits rank above all false labels' (so that any membership query can be answered by choosing a threshold), and models trained to minimize these losses perform better on class membership metrics. One of the new losses suggested, the batch cross-entropy, is particularly interesting in keeping with the recent work on using batch statistics; I would like to see this explored further (see below). The paper is well-written.

I am not sure of the relevance of this work as written. The authors discuss how related work (e.g. Grave et al.) scales computationally with K, which is undesirable; however, training the entire network with a non-CE objective function is an end-to-end model change, and practical uptake may suffer without further justification. The problem (and the proposed solution by changing training objective) is of interest because standard approaches ostensibly suffer unfavorable runtime-to-performance tradeoffs, so this should be demonstrated. I would be more comfortable if the authors actually evaluated runtime, preferably against one or two of the other heuristic baselines they cite.
The notation is a little uneven. The main idea is stated given the premise of Fig. 1, that there exist logits which are computed and passed through a softmax neuron, but this is never formally stated. (There are a few other very minor quibbles, e.g. top of pg. 6: sum should be over 1...k).

---

### Author Response · Authors · 2017-12-26
**A contribution to basic understanding of common building blocks**

We write this general comment as we believe an important aspect of our work was overlooked by the reviewers. We believe that basic understanding of fundamental building blocks of neural networks is a topic or high significance to ICLR. Neural network models already exist for a few decades and became very common over the past few years. Still, we lack basic understanding about the most common building blocks, such as the softmax function + cross entropy training mechanism and other common loss functions such as the binary cross entropy and NCE. Our work is concerned with understanding basic characteristics of the logits that result from a variety of the most common loss functions. We outline a simple property, and show that loss functions that optimize for this property at training time will yield logits at test time with characteristics that are much more suitable for some tasks. Moreover, we show that the most common softmax + cross entropy training mechanism is not optimizing for our proposed property. By doing so, we contribute to the basic understanding of the most common building blocks of neural network models.

---

### Decision · Program_Chairs · 2018-01-29
**ICLR 2018 Conference Acceptance Decision**

**Decision:**

Reject

**Comment:**

All of the reviewers have found some aspects of the formulation interesting, but they raised concerns regarding the practical use of the experimental setup.